# Evaluation of a Suicide Prevention Program for the Energy Sector

**DOI:** 10.3390/ijerph17176418

**Published:** 2020-09-03

**Authors:** Victoria Ross, Neil Caton, Sharna Mathieu, Jorgen Gullestrup, Kairi Kõlves

**Affiliations:** 1Australian Institute for Suicide Research and Prevention, School of Applied Psychology, Griffith University, Brisbane 4122, Australia; n.caton@griffith.edu.au (N.C.); s.mathieu@griffith.edu.au (S.M.); k.kolves@griffith.edu.au (K.K.); 2MATES in Construction, Astor Terrace, Spring Hill 4004, QLD, Australia; jorgen@micqld.org.au

**Keywords:** suicide prevention, training, evaluation, energy sector

## Abstract

There is evidence indicating that traditionally male-dominated occupations are associated with greater risk of suicide. In Australia, MATES in Construction was developed as an occupational health initiative to prevent suicides in the industry. The program has recently been applied to the energy industry; however, little is known regarding exposure to suicide and suicide prevention interventions in this sector. The study aimed to examine the effectiveness of MATES in Energy general awareness training (GAT), and estimate the prevalence of recent suicidal ideation and exposure to suicidal behaviors in workers. A before and after design was used to examine the effectiveness of GAT training. Data were collected from 4887 participants undertaking GAT training at energy sites across Queensland, Australia. In total, 2% (97) of participants reported recent suicidal thoughts, 65% of participants reported they had known someone who had attempted suicide, and 69% had known someone who died by suicide. Significant improvements were found on all suicide literacy items after GAT training. Younger people were more likely to be positively affected by the intervention. The results indicate that the MATES in Energy program is successfully transitioning from the construction industry, and offers the first empirically supported suicide intervention tailored to the energy sector.

## 1. Introduction

There is increasing evidence that indicates certain occupations are associated with a greater risk of suicide. In particular, workers in traditionally male-dominated industries (e.g., construction, agriculture, machinery) are at an increased risk of suicide compared to the general working population [1,2,3,4,5]. For example, in Australia, men in the construction, transport and agriculture industries are more likely to die by suicide than other occupations [1]. These males also experience poor mental well-being and suicidal ideation, e.g., [6], and a substantial portion report personally knowing someone who has died by suicide or who has had suicidal thoughts [7]. In addition to the devastating emotional impact of suicidal thoughts and behaviors for these industries, their workers and their families, there is a substantial economic burden borne by governments/healthcare systems and workplaces [8].

Research has identified characteristics at an individual (e.g., mental health, substance use, skill level), social (e.g., relationship breakdown, masculine norms, bullying) and occupational (e.g., casualness, financial uncertainty, work demands, strenuosity/fatigue) level that may serve as risk factors for suicide in these industries [3,9,10,11,12]. Furthermore, the lower mental health, poorer suicide literacy and increased stigma in these workplaces may serve as barriers to help-seeking [13]. Therefore, it has become increasingly important to develop and evaluate tailored workplace suicide prevention interventions, e.g., [14,15].

In Australia, MATES in Construction (MATES) was developed as a suicide prevention initiative in the building and construction industry [6]. The initiative involves a range of suicide prevention training programs, such as brief one-hour general awareness training (GAT), and other strategies such as case management provided to at-risk workers by trained MATES staff (i.e., typically social workers with specific industry experience of relatability) and mental health first-aid courses focused on suicide prevention [6]. Consistent with the MATES in construction model [6], the MATES in energy GAT consisted of suicide and mental health awareness training tailored to the energy industry, including the provision of information related to suicide warning signs and information designed to help de-stigmatize help-seeking/-offering among workers. Training is tiered so that those self-identified during GAT can go on to receive more in-depth Connector training to become volunteer worksite peer supporters [6]. Evidence suggests that the MATES programs are effective in increasing suicide literacy and help-seeking awareness in the construction and mining industries, and are viewed as acceptable by the intended workers and workplaces [6,8,16,17,18,19,20]. The program has been applied to the mining sector (Mates in Mining) [16,17], and in 2017 the MATES in Energy program was established to extend the MATES program to promote better mental health and reduce suicide within the energy sector. Under the MATES in Energy program, GAT has been thus far delivered to 7422 workers, Connector training to 1133, ASIST training to 99, and 89 workers have received case management.

To date, there has been little research into suicide and suicide prevention in the energy sector specifically, despite there being similar occupational characteristics here to the building and construction industries, and calls within the literature for occupation-specific suicide prevention initiatives e.g., [14,15]. Therefore, the current study aimed to explore the prevalence of recent suicidal ideation and exposure to suicidal behaviors in workers employed in the energy sector. This included whether participants have known someone who has attempted or died by suicide. Nevertheless, the main aim of the study was to examine the effectiveness of MATES in Energy GAT training.

## 2. Materials and Methods

### 2.1. Participants and Procedure

A before and after design was used to examine the effectiveness of GAT training. Data was collected from 4887 participants (Mean age = 42.64 ± 11.06 (SD); 73.2% males) undertaking GAT training between October 2017 and November 2018 at 49 energy sites across Queensland, Australia, where a short questionnaire was completed immediately before, and immediately after, 76 separate one-hour GAT training sessions. Participants were recruited from the overall energy industry, with occupations broadly comprising (excluding missing cases) the trades (e.g., electricians, laborers; *n* = 2016, 44.5%), managerial positions (*n* = 1000, 22.1%), administrative positions (e.g., receptionists, keyboard operators; *n* = 58, 1.3%) or other (*n* = 1457, 32.2%). MATES staff (i.e., field officers) administered the training and pre- and -post surveys, collected and entered the data, and provided the de-identified data file to the researchers for analysis. MATES field officers are paid employees, typically recruited from industry workers with lived experience of mental health and suicidality. Two field officers were assigned to complete the MATES in energy program (supported by other MATES field officers where required). MATES field officers undertake substantial in-house training in the delivery of GAT training over a 2-month period, are accredited Livingworks safeTALK trainers and have undergone Applied Suicide Intervention Skills Training (http://www.livingworks.com.au/). Training is delivered by two trainers with moderation and debriefing after each GAT delivery to ensure program integrity and quality control. Each trainer regularly partners with a supervisor trainer to ensure program fidelity. The study was approved by the Griffith University’s Human Research Ethics Committee (Reference number: 03/08/14586).

### 2.2. Survey Materials

Demographic information was collected for the date on which the intervention was undertaken, the training session ID, postal code, gender (1 = female, 2 = male) and age. The questionnaire contained items which asked whether the participant has known someone who had died by suicide, known someone who had attempted suicide, or whether they were currently experiencing suicidal thoughts themselves (currently or in the past two weeks). These were dichotomously answered as yes or no. There were seven items measuring suicide literacy, which were based upon those used in previous similar studies in the construction industry [7,13,19,20]. The measures used to assess suicide literacy were comprised of items assessing suicide awareness, knowledge, and attitudes to help-seeking and -giving. Each item was measured on a five-point Likert scale (1 = strongly disagree, 2 = disagree, 3 = neither agree nor disagree, 4 = agree, 5 = strongly agree), and one question measuring emotional well-being (1 = very poor, 2 = poor, 3 = OK, 4 = good, 5 = very good). The help-seeking item, *If I was going through a difficult time, feeling upset, or was thinking about suicide, I would be willing to seek help*, and a list of response options (e.g., close family, friend, doctor) were adapted from the General Help-Seeking Questionnaire [21]. Participants dichotomously indicated whether or not they would seek help from each option, where they were coded as “0” (i.e., did not intend to seek help) or “1” (i.e., intended to seek help). Finally, participants reported (answered as yes/no) whether they felt the GAT training was relevant for them, and whether they would recommend GAT training to others. Participants also rated the GAT training on a five-point Likert scale, where 1 = very poor, 2 = poor, 3 = OK, 4 = good and 5 = very good.

### 2.3. Data Analysis

Chi-square tests of association and independent samples t-tests were used to examine associations of gender and age with knowing someone who had attempted suicide or had died by suicide, or if they reported recently experiencing suicidal thoughts in the past week. Linear mixed-effects models were used to analyze the effectiveness of GAT for suicide literacy and emotional well-being. Logistic mixed models were used to analyze GAT’s effectiveness for help-seeking intentions, as the responses were recorded in a binary format. Both logistic and linear mixed models are statistical techniques that account for both within- and between-subjects variance, including the correlation between the repeated measures of participants. For the linear and logistic mixed models, time (pre- and post-intervention), date on which the intervention was undertaken, postal code, age, gender, whether they knew someone who had died by suicide, knew someone who had attempted suicide, or were recently experiencing suicidal thoughts, were entered as fixed effects. First-Order Autoregressive (AR1) and Unstructured (UN) covariance structures were examined using −2 Res Log Likelihood and Akaike’s Information Criterion (AIC), and both structures were applied to the levels of training session*person (as GAT was delivered in groups at different locations, participants were therefore nested within groups). The UN structure was identified as the model with the best fit with all dependent variables. Training session ID (the group that undertook the intervention together) was also included in the random intercept to model for between-group factors, and therefore account for potential variation in the delivery of the GAT program.

In line with previous research [19], we measured belief change for suicide awareness and emotional well-being by subtracting the post-test from the pre-test score (i.e., post-test minus pre-test) and including all five central variables (i.e., age, gender, whether they knew someone who had died by suicide, knew someone who had attempted suicide, or were currently experiencing suicidal thoughts) as predictor variables in the analysis. Given that help-seeking intentions were dichotomously answered, we measured belief change for these items as the interaction between time and each of the five predictor variables. Missing data occurred at rates between 0.00% and 2.11% for all but one variable. This exception was the age variable, for which 12.52% of the data were missing. Importantly, linear and logistic mixed models manage unbalanced data with the assumption that missing data in the outcome variable (time: before, after) are missing at random and are not excluded from the analyses; but missing data in the predictor variables are removed from the analyses via pairwise deletion. All statistical analyses were conducted using IBM SPSS version 26.0 (IBM, NY, USA).

## 3. Results

Overall, 2.03% (97 out of 4788, excluding missing cases) of the participants reported experiencing suicidal thoughts in the past week (or currently). Further, 65.38% (3137 out of 4798) participants reported that they had known someone who had attempted suicide, and 69.38% (3346 out of 4823) had known someone who died by suicide. Females were more likely to have known someone who had attempted suicide than males (*χ*^2^(1) = 15.92, *p* < 0.001, Cramer’s V = 0.06). However, there was no association between gender and knowing someone who had died by suicide (*χ*^2^(1) < 0.001, *p* = 0.99, Cramer’s V < 0.001) or reporting suicidal thoughts in the past week (or currently) (*χ*^2^(1) = 1.79, *p* = 0.18, Cramer’s V = 0.02). Those who had known someone (Mean age = 42.11 ± 11.12 (*SD*)) compared to those who had not known someone (Mean age = 43.74 ± 10.85 (SD)) who had attempted suicide were more likely to be younger (*t*(4203) = 4.51, *p* < 0.001, d = 0.15). Further, those who had reported having had suicidal thoughts in the past week (Mean age = 40.19 ± 11.53 (SD)) compared to those who did not (Mean age = 42.71 ± 11.04) were also more likely to be younger (*t*(4194) = 2.12, *p* = 0.03, d = 0.22). Conversely, those who had known someone (Mean age = 42.87 ± 11.08 (SD)) compared to those who had not known someone (Mean age = 42.11 ± 10.99 (SD)) who had died by suicide were more likely to be older (*t*(4224) = −2.03, *p* = 0.04, d = 0.07).

Linear mixed models (Table 1) showed a significant increase in the mean scores for six of the seven suicide literacy items, and a significant decrease for the first item, *Asking a workmate if they are having suicidal thoughts can increase his/her risk of suicide*, from before to after the GAT training. These results are illustrated in Figure 1. There was also a small improvement in how participants felt emotionally/mentally, from before to after the GAT training (Table 1). Logistic mixed models further revealed significant increases from pre- to post-intervention for all intended help-seeking sources, with the percentage of yes responses from pre- to post-intervention illustrated in Figure 2.

As previously mentioned, we also controlled for the five predictor variables when the linear and logistic mixed models for time were conducted. The relationships between these five predictor variables and suicide literacy, well-being and help-seeking intentions are presented in Table 2. Older individuals, males, and those who had not known someone who had attempted suicide were more likely to agree with the statement, “*Asking a workmate if they are having suicidal thoughts can increase his/her risk of suicide.*” Males and those who had not known someone who had died by suicide were more likely to agree with the statement, “*People considering suicide often send out warning signs or invitations*.” Younger individuals and those who had known someone who had attempted suicide were more likely to agree with the statement, “*Suicide is a serious problem in the energy industry.*” Females and those who were not currently experiencing suicidal thoughts were more likely to agree with the statement, “*If I was struggling with mental health issues, I would be willing to ask for help.*” Females, younger individuals, and those not currently experiencing suicidal thoughts were more likely to agree with the statement, “*If I was struggling with mental health issues, I would know who I would talk to, in order to get help*.” Females, younger individuals, and those who had known someone who had attempted suicide were more likely to agree with the statements, “*I would notice if a workmate was having a tough time and ask how he/she was doing*,” and “*If I knew a workmate was struggling then I would be willing to offer help*.” Younger individuals, and those who had not known someone who had attempted suicide or experienced suicidal thoughts, reported greater well-being.

Those who had not known someone who had attempted suicide were more likely to intend to seek help from a friend, workmate, supervisor, doctor, psychologist, counsellor or another source, but also no one; those who were experiencing suicidal thoughts were less likely to intend to seek help from a friend and close family member, more likely to seek help from a psychologist, but also more likely to intend to not seek help from anyone at all. Younger individuals were more likely to intend to seek help from a close family member, friend, workmate, psychologist or helpline, but older individuals were more likely to intend to seek help from a doctor or counsellor. Finally, females were more likely to intend to seek help from a friend, supervisor, doctor, psychologist or counsellor.

Linear mixed-effects models (see Table 3) also indicated that those who did not know someone who had died by suicide had a greater suicide literacy improvement, that is, disagreement with the statement *Asking a workmate if they are having suicidal thoughts can increase his/her risk of suicide* after the intervention compared to before. Females showed a greater positive change for the items *People considering suicide often send out warning signs or invitations* and *Suicide is a serious problem in the energy industry*, whereas males showed a greater positive change for the item *If I was struggling with mental health issues I would know who I would talk to, in order to get help*.

Given that gender predicted belief change for those three items, we also conducted sub-group analyses for the effect of time (before, after) on agreement with the statements *People considering suicide often send out warning signs or invitations, Suicide is a serious problem in the energy industry,* and *If I was struggling with mental health issues I would know who I would talk to, in order to get help,* as split by gender. Females, β = 0.48, *t*(11,033.36) = 30.68, *p* < 0.001, 95% CI[45, 51], showed greater agreement with the statement *People considering suicide often send out warning signs or invitations*, compared to males, β = 0.35, *t*(12,650.45) = 37.53, *p* < 0.001, 95% CI[33, 37], from before to after GAT training. Similarly, females, β = 0.35, *t*(11,032.97) = 24.63, *p* < 0.001, 95% CI[32, 38], showed greater agreement with the statement *Suicide is a serious problem in the energy industry*, compared to males, β = 0.29, *t*(12,651.86) = 32.26, *p* < 0.001, 95% CI[27, 31], from before to after GAT training. Conversely, males, β = 0.31, *t*(12,652.17) = 29.75, *p* < 0.001, 95% CI[29, 33], showed greater agreement with the statement *If I was struggling with mental health issues I would know who I would talk to, in order to get help*, compared to females, β = 0.26, *t*(11,033.28) = 17.44, *p* < 0.001, 95% CI[23, 29], from before to after GAT training

Further seen in Table 3, younger people were more likely to be positively affected by the intervention, such that they were more likely to disagree with the statement *Asking a workmate if they are having suicidal thoughts can increase his/her risk of suicide* after GAT. Younger people also showed greater belief change for the item *People considering suicide often send out warning signs of invitations*; but older people were more likely to show greater belief change for the item *I would notice if a workmate was having a tough time and ask how he/she was doing*. Finally, those who did not know someone who had attempted suicide showed greater improvement for the item *Suicide is a serious problem in the energy industry*. Logistic mixed models showed that none of the five predictor variables significantly predicted improvement across any of the help-seeking source items.

GAT was well received by participants. Overall, 94.04% (4500 out of 4785) of the participants felt that the GAT training was relevant for them, and 98.50% (4728 out of 4800) stated that they would recommend GAT training to others. Furthermore, 43.12% (2076 out of 4815) rated the GAT training as very good, 48.18% (2320) as good, 8.49% (409) as okay, and only 0.12% (6) rated the training as bad, and 0.08% (4) as very bad.

## 4. Discussion

The current study found that 2% of energy workers (particularly younger workers) reported experiencing current or recent suicidal ideation. A substantial portion of workers reported exposure to someone who had died by suicide or attempted suicide. These findings indicate that the introduction and subsequent evaluation of the MATES in Energy intervention has been timely. Results from the evaluation also suggest that the GAT training was acceptable to recipients, and had a positive impact on suicide literacy and help-seeking.

Importantly, the current findings regarding age differences in suicide literacy are consistent with those found by King and colleagues [19] in their earlier evaluation of GAT training in the construction industry. Just as in our study, younger workers reported poorer beliefs regarding the perceived adverse effect of discussing suicide and a lack of awareness of warning signs displayed by those contemplating suicide (this was particularly true for younger workers in manual roles, as compared to clerical/managerial roles [19]). Despite this, in our study, younger workers showed the biggest belief change from pre- to post-GAT in these domains. This is important, as having positive beliefs about discussing suicide and help-seeking is likely to work toward reducing stigma within the workplace, as well as helping assist those experiencing suicidal thoughts to seek help. The acceptability of GAT was supported by the training feedback, as nearly all participants rated the training highly, thought the GAT training was relevant, and would recommend it to others in their field.

Some gender differences were also seen, with females showing a greater positive change in several of the suicide awareness items and males showing greater improvement in one of the help-seeking items. These findings are important given that previous MATES evaluation studies focused on the male-dominated industries of construction and mining, both comprised of approximately 90% males [17,19,20]. In contrast, the current sample of energy workers was comprised of approximately 27% females. It will be important to further examine gender differences within the energy sector; specifically, whether GAT training could be effectively tailored by gender for maximum positive impact.

None of the five predictor variables significantly predicted improvement across the help-seeking source items. It is possible that the intervention presented help-seeking in a general way that did not prioritize certain groups over others, such that all groups were equally affected by the intervention as regards their intentions to seek help (e.g., young vs. old participants, males vs. females).

This study had several strengths. Previous research has only examined mortality rates among electrical workers, e.g., [22]. The current study utilized a large sample across manual, technical and managerial roles. This has broad implications for understanding the occupational health and wellbeing of workers in this industry in relation to experiences of suicidality and exposure to suicide. To the best of our knowledge, this is the first study to evaluate a workplace suicide prevention training program tailored for use within the energy industry, despite the impact of workplace interventions on reducing the economic burden of suicide on healthcare systems [8]. Our findings are consistent with previous similar research in the construction [6,7,13,18,19,20] and mining industries [16,17], whereby suicide literacy items, and willingness to both offer and seek help, significantly increased following GAT.

Limitations to the current study include the use of a convenience sample, self-report, before and after design, and the absence of a control group. This was the first study to apply a workplace suicide prevention intervention to the energy industry, and therefore used a simpler before-and-after training design that did not enable measurement of the long-term impact of training. Furthermore, self-report studies might be confounded by a response shift bias [23], and thus the alternative approach of a retrospective pre-test method [24] could instead be employed to control for this potential confound. Similarly, trainers also collected the confidential pre–post surveys from participants, and this may have inadvertently influenced responses. Nevertheless, the study methods are consistent with previous, similar research evaluating GAT suicide prevention in the construction industry, e.g., [19] and highlight the likely efficiency of a such a brief well-received workplace intervention. Additionally, the item used to assess sources of help-seeking referred to multiple sources of distress in the question, and while derived from a validated help-seeking instrument [21], it was not possible to determine whether help-seeking increased for workers going through a difficult time, feeling upset, and/or thinking about suicide. While there are procedures in place to mitigate site differences and maintain program fidelity (e.g., post-delivery moderation, paired trainers, extensive training), a further possible limitation may be site differences with regard to GAT delivery and uptake. Future research should include the collection of follow-up data and ensure that any possible site differences are measured and controlled for. While random assignment on the individual level might prove difficult in a real-world setting, future studies might benefit from the use of stepped wedge and/or cluster designs in workplaces [25], so as to further examine the effectiveness of the program, particularly in relation to any long-term reductions in actual suicide rates.

## 5. Conclusions

The results from the current study indicate that the MATES in Energy program is successfully transitioning from the construction industry, and offers the first empirically supported suicide intervention specifically tailored to the energy sector. Efficient workplace strategies that have an impact on suicide literacy and help-seeking avenues among workers are important. Overall, the MATES in Energy program was acceptable to workers and showed improvements in suicide literacy, particularly for younger workers, and these findings might prove informative for future suicide awareness programs for the energy sector.

## Figures and Tables

**Figure 1 ijerph-17-06418-f001:**
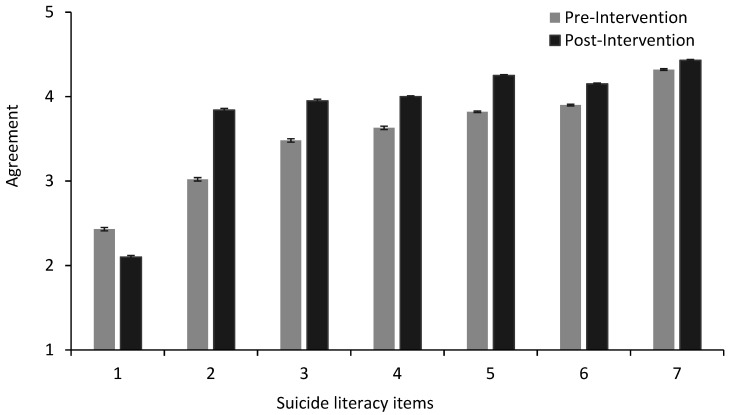
Estimated marginal means for suicide literacy pre- and post-intervention. *Note.* Error bars represent standard errors. On the *x*-axis, 1 = “Asking a workmate if they are having suicidal thoughts can increase his/her risk of suicide”, 2 = “People considering suicide often send out warning signs or invitations”, 3 = “Suicide is a serious problem in the energy industry”, 4 = “If I was struggling with mental health issues, I would be willing to ask for help”, 5 = “If I was struggling with mental health issues I would know who I would talk to, in order to get help”, 6 = “I would notice if a workmate was having a tough time and ask how he/she was doing”, 7 = “If I knew a workmate was struggling then I would be willing to offer help”.

**Figure 2 ijerph-17-06418-f002:**
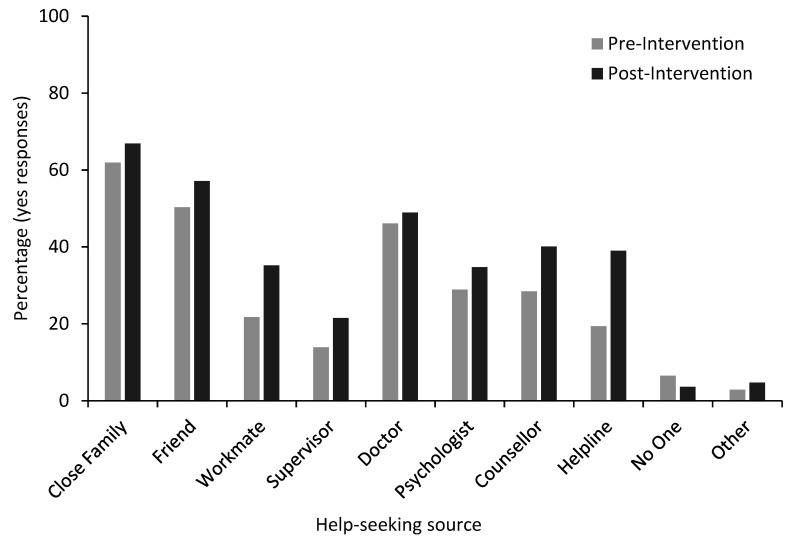
Percentages of help-seeking intentions by source (yes responses) pre- and post-intervention.

**Table 1 ijerph-17-06418-t001:** Descriptive statistics and fixed effect estimates for pre–post-GAT training from the linear mixed-effects regression.

Item	Pre		Post				
**Suicide literacy**	***M*^1^**	***SE***	***M*^1^**	***SE***	***β***	***t***	***p***
“Asking a workmate if they are having suicidal thoughts can increase his/her risk of suicide”	2.43	0.02	2.10	0.02	−0.19	−24.29	<0.001
“People considering suicide often send out warning signs or invitations”	3.02	0.02	3.84	0.02	0.39	47.86	<0.001
“Suicide is a serious problem in the energy industry”	3.48	0.02	3.95	0.02	0.30	40.33	<0.001
“If I was struggling with mental health issues, I would be willing to ask for help”	3.63	0.02	4.00	0.01	0.23	31.34	<0.001
“If I was struggling with mental health issues, I would know who I would talk to, in order to get help”	3.82	0.01	4.25	0.01	0.30	34.46	<0.001
“I would notice if a workmate was having a tough time and ask how he/she was doing”	3.90	0.01	4.15	0.01	0.20	22.87	<0.001
“If I knew a workmate was struggling then I would be willing to offer help”	4.32	0.01	4.43	0.01	0.10	12.98	<0.001
**Well-being**							
“So far today, the best way to describe how I’m feeling emotionally/mentally is…”	3.82	0.01	3.86	0.01	0.02	5.95	<0.001
**Help-seeking source (yes responses)**	***N***	**%**	***N***	**%**	***OR*^2^**	***t***	***p***
Close family	3027	61.94	3269	66.89	1.27	4.84	<0.001
Friend	2458	50.30	2791	57.11	1.36	6.34	<0.001
Workmate	1059	21.67	1718	35.15	2.01	13.44	<0.001
A supervisor	678	13.87	1051	21.51	1.74	9.24	<0.001
My doctor	2253	46.10	2388	48.86	1.13	2.44	0.01
Psychologist	1414	28.93	1697	34.72	1.33	5.62	<0.001
Counsellor	1389	28.42	1960	40.11	1.72	10.89	<0.001
A helpline	947	19.38	1907	39.02	2.74	18.76	<0.001
No one	317	6.49	174	3.56	0.56	−5.05	<0.001
Other	140	2.86	228	4.67	2.01	5.33	<0.001

Note. ^1^ estimated marginal mean. ^2^
*OR* = odds ratio.

**Table 2 ijerph-17-06418-t002:** Fixed effect estimates for the predictors of suicide awareness, well-being and help-seeking scores.

Item	Died by Suicide ^1^		Suicide Attempt ^2^		Suicidal Thoughts ^3^		Age		Gender	
**Suicide literacy**	***β***	***p***	***β***	***p***	***β***	***p***	***β***	***p***	***β***	***p***
“Asking a workmate if they are having suicidal thoughts can increase his/her risk of suicide”	0.03	0.09	−0.05	0.001	0.03	0.052	0.04	0.001	0.11	<0.001
“People considering suicide often send out warning signs or invitations”	−0.05	<0.001	0.01	0.59	0.00	0.78	0.01	0.26	0.05	<0.001
“Suicide is a serious problem in the energy industry”	0.01	0.64	0.08	<0.001	0.01	0.56	−0.05	<0.001	−0.03	0.06
“If I was struggling with mental health issues, I would be willing to ask for help”	0.00	0.90	−0.01	0.39	−0.08	<0.001	0.00	0.92	−0.11	<0.001
“If I was struggling with mental health issues, I would know who I would talk to, in order to get help”	0.00	0.94	0.00	0.90	−0.05	<0.001	−0.04	<0.001	−0.11	<0.001
“I would notice if a workmate was having a tough time and ask how he/she was doing”	0.00	0.94	0.04	0.005	0.01	0.65	−0.05	<0.001	−0.17	<0.001
“If I knew a workmate was struggling then I would be willing to offer help”	0.03	0.08	0.06	<0.001	−0.02	0.10	−0.10	<0.001	−0.17	<0.001
**Well-being**										
“So far today, the best way to describe how I’m feeling emotionally/mentally is…”	0.00	0.84	−0.06	<0.001	−0.17	<0.001	−0.04	0.002	−0.01	0.44
**Help-seeking source**	^4^ *OR*	*p*	^4^ *OR*	*p*	^4^ *OR*	*p*	^4^ *OR*	*p*	^4^ *OR*	*p*
Close family	0.98	0.32	1.00	0.96	0.79	<0.001	0.97	<0.001	0.94	0.33
Friend	1.01	0.69	1.05	<0.001	0.80	<0.001	0.96	<0.001	0.58	<0.001
Workmate	1.00	0.84	1.03	0.046	0.96	0.35	0.99	<0.001	1.08	0.20
A supervisor	0.97	0.13	1.05	0.01	1.04	0.40	1.00	0.17	0.82	0.004
My doctor	1.00	0.84	1.03	0.02	1.01	0.82	1.02	<0.001	0.47	<0.001
Psychologist	0.99	0.71	1.06	<0.001	1.11	0.01	0.99	<0.001	0.55	<0.001
Counsellor	0.98	0.11	1.05	<0.001	1.02	0.56	1.01	<0.001	0.75	<0.001
A helpline	0.99	0.48	0.99	0.73	0.99	0.90	0.99	0.02	0.91	0.12
No one	1.03	0.35	1.07	0.07	1.51	<0.001	1.01	0.11	1.23	0.14
Other	1.02	0.57	1.15	<0.001	1.13	0.14	0.99	0.27	1.02	0.90

Note. ^1^ knows someone who has died by suicide; ^2^ knows someone who has attempted suicide; ^3^ currently, or in the last week, has had suicidal thoughts. ^4^
*OR* = odds ratio.

**Table 3 ijerph-17-06418-t003:** Fixed effect estimates for the predictors of change for GAT from the linear mixed effects models.

Item	Died by Suicide ^1^		Suicide Atempt ^2^		Suicidal Thoughts ^3^		Age		Gender	
**Suicide awareness**	***t***	***p***	***t***	***p***	***t***	***p***	***t***	***p***	***t***	***p***
“Asking a workmate if they are having suicidal thoughts can increase his/her risk of suicide”	2.10	0.04	−0.20	0.98	−0.64	0.53	2.10	0.04	−0.86	0.39
“People considering suicide often send out warning signs or invitations”	0.61	0.54	−0.70	0.49	0.37	0.71	−2.31	0.02	−6.81	<0.001
“Suicide is a serious problem in the energy industry”	−0.87	0.38	−3.42	0.001	−1.49	0.14	−0.12	0.82	−3.70	<0.001
“If I was struggling with mental health issues, I would be willing to ask for help”	−1.30	0.19	0.79	0.43	−0.35	0.73	−1.59	0.11	−0.02	0.98
“If I was struggling with mental health issues, I would know who I would talk to, in order to get help”	−1.21	0.23	−0.48	0.63	0.69	0.49	−0.24	0.81	2.51	0.01
“I would notice if a workmate was having a tough time and ask how he/she was doing”	−0.73	0.46	0.98	0.33	1.02	0.31	2.95	0.003	−0.16	0.87
“If I knew a workmate was struggling then I would be willing to offer help”	−1.12	0.26	−0.93	0.35	0.44	0.66	0.04	0.97	1.05	0.29
**Well-being**										
“So far today, the best way to describe how I’m feeling emotionally/mentally is…”	−0.50	0.62	−0.76	0.45	1.01	0.31	1.65	0.10	−0.34	0.75
**Help-seeking source**										
Close family	−0.39	0.70	0.47	0.64	0.36	0.72	0.94	0.34	0.14	0.88
Friend	0.22	0.82	−0.12	0.90	−0.09	0.93	0.35	0.72	0.19	0.85
Workmate	−1.28	0.20	−0.03	0.97	−0.74	0.45	0.56	0.58	0.22	0.83
A supervisor	−0.28	0.77	−0.59	0.55	−1.13	0.26	−0.29	0.77	0.29	0.77
My doctor	−1.32	0.19	−0.20	0.84	0.15	0.87	−0.06	0.94	−0.41	0.68
Psychologist	−0.11	0.91	−0.15	0.87	−0.86	0.39	0.45	0.65	−1.01	0.31
Counsellor	0.40	0.71	−0.32	0.74	0.84	0.40	0.84	0.40	0.19	0.85
A helpline	−0.07	0.94	−0.77	0.44	−1.46	0.15	−1.14	0.26	1.40	0.16
No one	0.17	0.86	−0.32	0.75	1.00	0.31	−0.28	0.78	1.27	0.20
Other	1.10	0.27	−0.68	0.49	−1.09	0.27	−0.46	0.64	−0.45	0.65

Note. ^1^ knows someone who has died by suicide; ^2^ knows someone who has attempted suicide; ^3^ currently, or in the last week, has had suicidal thoughts.

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
