# Peer review of "Evaluation of a Suicide Prevention Program for the Energy Sector"

_ijerph, 2020, doi:10.3390/ijerph17176418_

Round 1

Reviewer 1 Report

The authors tracked suicide-related cognitive changes in Australia's energy sector workers through the GAT.

Major Issues

1) Within the framework of the methodology chosen by the authors, the scientifically rational and appropriate analytical methodology is selected.

2) However, the most important endpoint in a suicide-related intervention program is a decrease in suicide rate, which is very difficult to measure in specific research. Therefore, the effectiveness presented in this study is a kind of soft outcome

3) In addition, although this study deals with interventions for workers in the energy sector, the basic characteristics of workers in the energy sector are insufficient.

4) This study is very well documented, except for concerns about the fundamental themes and values of these studies. According to line 197, the authors seem to have already fully understood this. Therefore, a more technical approach to these themes and values should be presented.

Minor Issues

1) There is a lack of sufficient explanation of the concept of GAT.

2) According to Table 2, the results of this multi-variable analysis seem to be sensitive to age and gender. Therefore, it would be helpful to present the basic characteristics of workers.

3) Subgroup analysis can also be considered, especially if it is affected by gender.

Author Response

Responses to the reviewer have been uploaded as an attachment.

Reviewer 2 Report

This is an important and novel piece of work evaluating the effectiveness of a suicide prevention programme in the energy industry. Pending some clarifications in methodology/results, formatting amendments, and rechecking of reported values, it would make an excellent contribution to the field of suicide prevention. I have detailed my comments below regarding each section of the manuscript.

Introduction

p 2 Line 48. It would be helpful to give more details about the content and activities of the awareness training and the other strategies (case management and mental health first aid) used. I understand there is probably a lot of information to fit in but I think it would be important for the readers to understand what the intervention actually involved. If the article is pushed for space, can the authors include the details in a supplementary section?

Method and Materials

The method section would benefit from sub-sectioning for clearer structure. e.g. Participants, Design, Materials, Procedure, Data analysis etc

p. 2 Line 68. and throughout the report. I am assuming “Mage” means “Mean Age”, can this be amended throughout the report as it doesn’t read right as one word. Also, is +11.06 the Standard Deviation (SD) or Stand Error (SE)? Please denote the appropriate one and amend throughout the report.

p.2 Line 69. How many sites were involved?

p. 2 Line 74. Can the authors give more details in the report of who the MATES staff were? Were they paid employees or volunteers? How many staff is involved in the programme delivery? What training have they received to deliver the programme? What steps have been taken for quality control and ensuring consistency in delivery? 

p.2 Line 74. Were the same MATE staff delivering the training also collecting the survey data? If that was the case, what steps were taken to mitigate bias in participants rating the programme favourably as a result of demand characteristics imposed by the people delivering the programme? This is also a relevant point to raise in the limitation section of the discussion.

p.2 Line 84. “five-point Likert scale from 1=strongly disagree to 5=strongly agree” - can all the options be listed?

Results

p.3 Line 120. First sentence of the results is a repeat of line 67-68 in the previous page.

How much missing data is there? Are all 4887 cases included in the analysis? In page 2 line 72, there is mentions of missing data. Can the authors include details of how much data is missing and how is this dealt with in the analyses?

p.3 Line 120. “Two percent (4,691)” - I think there is a mistake here, 2% of 4887 is not 4691. 

p.3 Line 121. 3137 of 4887 is not 65% but slightly lower (64.19071005…); as .19 is less than .5 this should be round rounded down to 64, not rounded up. Can the authors go back to recheck all values reported are rounded correctly. 

A related point - there are inconsistencies in the way values are reported throughout the report. For example, line 120, 73.2% is reported to 3 significant figures,  whereas line 121, 65% is rounded to 2 significant figures only. My suggestion would be to have precision of 3 significant values for everything reported - same goes for t/p values, make it consistent throughout.

For all statistical tests, please report the effect sizes, e.g. for chi-square, report Cramer’s V and for t-tests, report Cohen’s d.

t and p is italized in the first row of Table 1 but not further down in the Help-seeking source section and also in the text throughout the report. Please can the authors go back to check formatting of the report (e.g. M, SE, N, n, t, p should be in italic).

Table 1. Can β be reported for the linear mixed models?

The authors mentioned previously in the method section that a number of variables were entered into the Linear Mixed models, however, in Table 1 and in text only the fixed effects of time were reported and discussed. Can the results of the full model outputs be summarised (maybe in a supplementary section)? 

Paragraph beginning Line 176. As mentioned before, the percentages seem to be slightly out. For example,4500/4887*100 =92.08…. not 94%; next, 4728/4887*100 = 96.74… not 99%. I would strongly advise going rechecking all values throughout the report carefully to ensure calculations are correct. If there are missing data, these need to be noted, and the actual sample size used for calculating these %s should be reported.

Author Response

(The authors gave the same response as above.)

Reviewer 3 Report

Abstract: What does the 4,691 after the two percent represent signify? The number is too big to represent 2% of the sample (See line 120 in the results for the same concern).

Lines 42-43: Please clarify if the risk factors are for suicide.

Lines 50-51: How are the individuals identified during GAT selected to receive the in-depth connector training for per support? Do they need to meet a certain set of requirements, have “extra” interest in becoming a peer supporter, or something different? Please clarify.

Please add subheadings to the methods (e.g., study design, participants and recruitment, measures, etc.). It is somewhat difficult to follow in its current form.

Who administered the 76 GAT trainings? Was it the same person or group of people and how are they trained? Variation in delivery could skew the results.

The item assessing help-seeking intentions by source is triple barreled. Participants are asked to select ‘yes’ or ‘no’ in response to whether or not they would seek help if they were 1) going through a difficult time, 2) feeling upset, 3) or thinking about suicide. Findings related to help-seeking source apply to participant’s intentions to seek help (and from whom) when they are going through a hard time, feeling upset, and thinking about suicide, not simply suicide. This is a methodological limitation with implications for the evaluation of the program – it’s impossible to determine whether participants are willing to seek help for having hard time, feeling upset, or thinking about suicide, regardless of source. At a minimum, this needs to be addressed as a limitation in the paper.

Which of the seven items measure suicide awareness, knowledge, help-seeking, and giving? If distinguishing between those items in the methods, it would be helpful to see them listed as sub scales in the results.  

What was the rationale for labeling the suicide awareness items? They appear to be modified from the suicide literacy scale used in the King et al., 2019 article. I’m curious why these weren’t labeled as suicide literacy? At a minimum, these items should be described as a modified version of the suicide literacy measure and cited.  

Line 120: What does the 4,691 after the two percent represent signify? The number is too big so represent 2% of the sample.  

The discussion is somewhat disjointed in its current format. Initially, I thought the discussion of findings was limited to the first paragraph. However, I found the rest of the discussion after the strengths and limitations. I recommend discussing all the results first and concluding that section with the paragraphs on the strengths and limitations. In other words, move paragraphs four and five (discussion of findings) ahead of paragraphs two and three (strengths and limitations) in this section.

Please add to the discussion why you believe the five predictor variables did not predict improvement across any of the help-seeking source items.

Potential variation in the delivery of the GAT program should be discussed in the limitations.

Author Response

(The authors gave the same response as above.)

Round 2

Reviewer 1 Report

Most of the problems I raised have been resolved.
In particular, I am fortunate that through sufficient explanations, I can understand the value of research.

However, the fact that the research subject itself is difficult to generalize is considered a limitation of the research itself.

Author Response

Responses have been uploaded.

Reviewer 3 Report

Thank you for responding and addressing to the concerns about the paper. I have two additional suggestions for improving the consistency and clarity of the paper.

Lines 100-101: It is important readers understand that suicide literacy is not comprised of four subscales. For clarity, please revise the sentence describing the items measuring suicide literacy from: “The items assessed suicide awareness, and knowledge, and attitudes to help-seeking and giving,” to “The measures used to assess suicide literacy is comprised of items assessing suicide awareness, knowledge, and attitudes to help-seeking and giving.”

Line 151-152: Please add “(or currently)” to the end of the sentence but before the Chi Square in line 152. Line should read, “…or reporting suicidal thoughts in the past week (or currently).”

Author Response

The responses have been uploaded.
